# The Association between Vegan Dietary Patterns and Physical Activity—A Cross-Sectional Online Survey

**DOI:** 10.3390/nu15081847

**Published:** 2023-04-12

**Authors:** Sandra Haider, Alina Sima, Tilman Kühn, Maria Wakolbinger

**Affiliations:** 1Department of Social and Preventive Medicine, Center for Public Health, Medical University of Vienna, Kinderspitalgasse 15/1, 1090 Vienna, Austria; sima.alina99@gmail.com (A.S.); maria.wakolbinger@meduniwien.ac.at (M.W.); 2Center for Public Health, Medical University of Vienna, Kinderspitalgasse 15/1, 1090 Vienna, Austria; tilman.kuehn@meduniwien.ac.at; 3Department of Nutritional Sciences, University of Vienna, Josef-Holaubek-Platz 2, 1090 Vienna, Austria

**Keywords:** vegan, physical activity, dietary pattern, strength training, yoga, aerobic physical activity

## Abstract

A balanced diet and sufficient physical activity (PA) are known to have positive health effects. The relationship between a vegan diet and PA levels is understudied. This cross-sectional online survey aimed to analyze whether different vegan dietary patterns differ in PA. In total, 516 vegan participants were included (June to August 2022). Different dietary patterns were compiled through principal component analysis, while group differences were calculated using independent tests, or chi-squared tests as well as logistic regression analyses. The population had an average age of 28.0 (SD: 7.7) years and had been living vegan for 2.6 (95% CI: 2.5–3.0) years. Two dietary patterns, the “convenience” and the “health-conscious” group, were identified. People with a convenience dietary pattern had significantly higher odds of sitting more (OR 1.10, 95% CI 1.04–1.18) and not achieving aerobic PA (OR 1.81, 95% CI 1.18–2.79) or strength training recommendations (OR 1.81, 95% CI 1.26–2.61) than people with a health-conscious dietary pattern. This study suggests the heterogeneity of vegan diets and that dietary patterns must be differentiated, as they also differ in the level of PA. Additional studies involving complete dietary assessment with a focus on ultraprocessed foods, blood metabolite analysis, and objective PA assessment are required.

## 1. Introduction

Veganism is a type of vegetarianism or plant-based diet with the most stringent restrictions, excluding all animal-source foods and by-products [1]. Especially in high-income countries, the vegan population is increasing due to its popularity [2]. In Austria, for example, in 2017 the proportion of vegetarians and vegans was 6%, and at the beginning of 2021, it was around 11%. Overall, Austria’s population share of vegans is 2% [3]. At the same time, Europe’s market for meat and dairy substitutes (vegan meat and dairy alternatives) is worth 1.7 billion euros [4].

In general, a plant-based diet low in salt, saturated fats, and added sugars is recommended as part of a healthy lifestyle. Indeed, such diets have been linked to a lower risk of premature mortality and protection against noncommunicable diseases (NCDs) [5]. In order to provide science-based diet recommendations for vegans, the German Giessen Vegan Food Pyramid was developed in 2018 [6]. The base of the pyramid comprises nonalcoholic and unsweetened beverages, vegetables, fruits, whole grains, and potatoes. Further up the pyramid are nuts and seeds, pulses and other protein-rich foods, nondairy milk alternatives, and plant oils and fats. Additional recommendations are daily vitamin B12 supplementation, iodized salt or sea salt enriched with iodine-containing algae, and outdoor activity for vitamin D synthesis [6].

Despite the health benefits of a plant-based diet, not all can be considered healthy. For the most part, plant-based diets are associated with healthful and minimally processed plant foods [5]. On the other hand, refined grains, sugar-sweetened beverages, snacks, and confectionery can be considered “plant-based”, as they or their ingredients are derived from plants, but are to be classified as ultraprocessed foods (UPFs) [5]. Therefore, UPFs may also be found in modern plant-based diets or veganism [7]. This consumption of UPFs is associated with higher all-cause mortality, as indicated by an analysis of the health-conscious US Adventist population, with many vegetarians [8]. A cross-sectional study from the NutriNet-Santé cohort found that not all vegetarian diets are necessarily healthy due to the potential negative effects of UPFs on nutritional quality, as vegans had the highest consumption of UPFs compared to meat eaters, pesco-vegetarians, and vegetarians [9]. A study by Gallagher et al. [10] further demonstrated that vegan diets vary substantially in diet quality, with some vegan diets incorporating a range of food groups, representing a traditional well-planned vegan diet, and other vegan diets consisting of large quantities of processed and UPFs [10]. According to a cross-sectional survey of Brazilian vegetarians and vegans [11], 41% of vegans drank sugary beverages daily (including 100% fruit juice). Additionally, it discovered a link between eating UPFs and a greater risk of overweight in vegans and vegetarians [11]. Overall, few published studies have addressed the aspect of vegan diet heterogeneity, but those suggest that vegan diet quality varies significantly.

Both a balanced diet and sufficient physical activity (PA) have positive health benefits, and they contribute to preventing and managing NCDs [12,13]. In that regard, vegan dietary patterns, both healthy and unhealthy, have yet to be linked to PA. According to the international and Austrian PA guidelines, adults should engage in at least 150–300 min of moderate-intensity aerobic PA, 75–150 min of vigorous-intensity aerobic PA, or an equivalent combination of moderate- and vigorous-intensity activity throughout the week [14,15]. In addition, muscle-strengthening exercises at moderate or greater intensity should be performed on ≥2 days/week, targeting all major muscle groups [14,15]. Time spent being sedentary should be limited.

Due to the heterogeneity in vegan diets and as studies on the association between vegan dietary patterns and PA are broadly missing, the aim of this study was to (1) assess dietary patterns in people who have adopted a vegan diet, and (2) analyze the association between different vegan dietary patterns and the extent of PA.

## 2. Materials and Methods

### 2.1. Study Design

This examination was a cross-sectional online survey. People were asked to complete the questionnaire between June and August 2022, using the software package Sosci Survey (www.soscisurvey.de (accessed on 2 May 2022); Munich, Germany). A sample size of more than 60 participants was aimed at as the minimum for factor analysis [16,17].

### 2.2. Participants, Recruitment

The recruitment took place via Facebook, including the groups “Vienna Vegan” and “Austria Vegan”, and the Instagram accounts “vegans_vie”, “thelalavienna” and “veganfindsaustria”. These groups were chosen based on the recommendations of vegans, who were active on these platforms. The link was also sent to the students of the Medical University of Vienna via Facebook and Telegram. Furthermore, colleagues from the Research Institute for Plant-Based Nutrition in Germany sent out the survey in their newsletter, and participants were recruited via Vegane Gesellschaft Österreich.

The following inclusion criteria were applied: (1) persons ≥ 18 years of age, (2) vegan diet for >3 months, (3) German-speaking persons, (4) signed consent form, and (5) ≥90% of the questionnaire completed. Exclusion criteria were (1) an omnivorous diet and (2) lacto-ovo-vegetarians, ovo-vegetarians, and lacto-vegetarians.

### 2.3. Survey Methods

Population characteristics such as time living vegan (years), age, gender (female, male, nonbinary or not specified), education (compulsory school, A-level, university), employment (students or in training, employee or self-employed, other), and smoking status (yes, occasionally, former, never) were gathered, as well as body weight (to the nearest kilo) and height (accurate to cm). Body mass index (BMI) was calculated and categorized into underweight (<18.5 kg/m^2^), normal weight (18.5–<25.0 kg/m^2^), overweight (25.0–<30.0 kg/m^2^), and obesity (≥30 kg/m^2^) [18].

A vegan diet was assessed with the English-validated Food Frequency Questionnaire (FFQ) from the EPIC-Norfolk Questionnaire [19]. The basis of the German FFQ was the adapted version from Gallagher et al. [10] and modified to include questions representative of foods and drinks suitable for vegans. We then modified it using country-specific foods. The questionnaire asked for dietary habits over the past month and the frequency of consumption for each food was coded as follows: ≤1/month, 1–3/month, ≤1/week, 2–4/week, 5–6/week, ≤1/day, 2–3/day, 4–5/day, ≥6/day [10].

Reasons for a vegan diet were collected with the question “For what reasons did you decide on a vegan diet?” with multiple response options, including (1) health aspects, (2) environmental protection, (3) animal welfare, (4) religious beliefs, (5) weight reduction and (6) others. Supplementation (yes or no) of vitamin B12, vitamin D, omega-3 fatty acids, iodine, iron, calcium, zinc, potassium, vitamin K2, and selenium was asked about, with multiple response options. Additionally, personal opinions on diet quality were gathered with the questions “Do you feel that your eating habits are balanced?”, “Do you feel that you spend more or less money on food because of your vegan diet?” and “How often do you eat in vegan-friendly cafés or restaurants?”.

Physical activity was assessed with parts of the Global Physical Activity Questionnaire (GPAQ) [20]. In detail, the GPAQ recorded sitting time on a typical day (hours/day). Additionally, minutes of moderate and vigorous aerobic PA during leisure time (min/week) were asked with the questions “Do you engage in moderate PA or sport in your free time, where breathing and pulse rate increase slightly?” and “On a 10-item scale, you would rate these activities as 5–6?” Furthermore, it was asked “How many days a week do you invest in moderate PA?” and “How much time do you invest in moderate PA per week?” The same was done for vigorous PA. To calculate if aerobic PA recommendations were fulfilled, the vigorous minutes were counted twice and added to the moderate minutes [14,15]. Frequency of strength training (days/week) was asked about. If people did ≥2 days/week, PA recommendations were fulfilled [14,15]. Based on the GPAQ Analysis Guide, participants with implausible data were excluded [20]. Additionally, five questions about yoga were added. Based on the study by Neumark-Sztainer and colleagues [21], participants were asked (1) Do you practice yoga regularly? (yes, no); (2) How often do you practice yoga? (1–2/week, 3–4×/week, 5–6×/week, every day); (3) How would you rate your level? (beginner, advanced, professional, yoga teacher); (4) Which style do you prefer? (hatha, yin, ashtanga, nidra, vinyasa, other); and (5) Which positive effects are most important for you? (strength building, flexibility, stress reduction, pain reduction, health care, weight reduction).

### 2.4. Ethical Considerations

The study was conducted according to the Declaration of Helsinki. Ethics approval was gained from the Medical University of Vienna (EK 1212/2022; 9 June 2022). All participants provided written informed consent in advance of participation.

### 2.5. Statistics

For deriving dietary patterns, we have used an a posteriori and a priori approach. First, food and beverage items were combined and collapsed into 18 food groups based on Gallagher et al. [10] for applying principal component analysis (PCA; a posteriori approach). Both Bartlett’s test (chi-squared = 1961.4, *p* < 0.001) and the Kaiser–Meyer–Olkin Measure of Sampling Adequacy (KMO = 0.805) indicated that the variables were suitable for PCA. Therefore, the dietary patterns were assembled from the 18 main food groups and eating behavior variables by PCA using varimax rotation [22]. The extraction of the number of components was determined by applying the following criteria: value >1, identification of a break in the scree plot, and interpretability of the components. Although it indicates the presence of five factors with eigenvalues > 1.0, based on the scree plot and theoretical considerations, a two-factor solution was chosen, which explained 33.8% of the variance. Food items with absolute factor loading ≥ 0.3 accounted for each component (Table 1).

Food groups were not included if they had an absolute loading <0.3. The cross-loadings found indicate that a third factor cannot be clearly separated from the other two. Based on prior content knowledge, this solution can nevertheless be assumed. Thus, two factors are present: “convenience dietary pattern” and “health-conscious dietary pattern”. The factor score coefficients were calculated by summing the amount of daily frequency for each food group and were weighted by the loading factor determined by PCA. Each participant was given a factor score coefficient for each defined dietary pattern.

In addition to data-driven dietary patterns, we assessed diet quality by the modified plant-based diet index (PDI; a priori approach) [23]. For the PDI, we used the 18 food groups divided into healthy plant-based foods and dietary behavior (cooking with fresh ingredients, vegetables, protein alternatives, creating own recipes, fruits, dairy alternatives, potatoes, whole grains, and vegetable oils and fats) and less healthy plant-based foods and dietary behavior (processed fish and meat alternatives, vegan savory snacks, vegan processed foods, vegan sauces and condiments, vegan cakes and biscuits, fruit juices/smoothies, vegan sweets and desserts, refined grains, and vegan convenience meals and snacks). Following Satija et al. [24], the variables (healthy and less healthy plant-based foods) were classified into quintiles and assigned positive for the healthful plant-based diet index (hPDI) or reverse scores for the unhealthful plant-based diet index (uPDI). Participants above the highest quintile of a healthy plant-based food group received a positive score of 5, while those below the lowest quintile received a score of 1. This scoring pattern was reversed when applied to the less healthy plant-based food groups, e.g., the highest quintile received a score of 1 and the lowest quintile a score of 5. With this analysis, we divided the people into three groups: 1st tertile = convenience; 2nd tertile = traditional; 3rd tertile = health-conscious dietary pattern. The results of the a priori method are shown in the Appendix A.

To analyze differences between the vegan dietary pattern groups, independent *t*-tests, Mann–Whitney-U-test or ANOVAs were calculated for metric data. For categorical data, chi-squared tests were used. Finally, logistic regression analyses were used to investigate the association between the vegan dietary pattern groups and PA. These analyses were adjusted for gender (female, male, nonbinary/unspecified), employment (students or in training, employee or self-employed, other), smoking status (yes, occasionally, former, never), and time being vegan (years, continuous), as these parameters were shown to be significantly different between the two dietary patterns. To interpret the goodness of fit, we presented the Nagelkerke R^2^. All calculations were done in IBM^®^ SPSS^®^ Statistics version 27 (Armonk, NY, USA), with *p*-values < 0.05 considered statistically significant and all tests two-sided.

## 3. Results

A total of 537 people filled out the questionnaire. Twenty participants did not meet the inclusion criteria, and one participant was excluded due to implausible PA data. This resulted in a sample size of 516 respondents. Of these, no dietary pattern could be calculated for two people due to incomplete dietary data.

### 3.1. Characteristics

The included participants had a mean age (SD) of 28.0 (7.7) years, and had been vegan for a median time of 2.6 (95% CI: 2.5–3.0) years. In sum, 85% were female and 91% had at least an A-level education. A majority (56%) were employees or self-employed, 41% were students or in training, 7% were smokers, 79% classified as normal weight, 15% as overweight or obese, and 6% as underweight.

The results of the PCA allocated the participants into convenience and health-conscious dietary patterns (Table 2). The convenience dietary pattern is characterized by higher consumption of processed fish and meat alternatives, vegan savory snacks, processed foods, sauces and condiments, cakes and biscuits, sweets and desserts, convenience meals and snacks, fruit juices/smoothies, and refined grains. The health-conscious dietary pattern is characterized by higher consumption of vegetables, fruits, protein alternatives (e.g., tofu), dairy alternatives, potatoes, whole grains, vegetable oils and fats, and cooking with fresh ingredients and creating own recipes. People in the health-conscious group had been living vegan for a significantly longer time. We found significantly more female participants, more students, and nonsmokers in this dietary pattern (Table 2). No significant differences between groups were found in age, education, or BMI.

Results of the dietary patterns based on the modified PDI score (Appendix A) also showed that significantly more women were in the health-conscious group (*p* = 0.048). In this analysis, there was a significant difference in BMI: people in the health-conscious group had significantly lower BMI than those in the convenience dietary pattern (*p* = 0.016).

### 3.2. Vegan Diet

Reason for adopting a vegan diet was in 91% due to animal welfare, in 73% due to environmental protection, and in 60% due to health aspects. A significant group difference between the convenience and the health-conscious dietary patterns was found, whereby animal welfare played a greater role in the convenience group (Table 3). This significant difference could not be found in the dietary patterns based on the modified PDI score (Appendix A).

Concerning supplements, 94% of all participants took vitamin B12, 75% vitamin D, 48% iron, and 43% omega-3 fatty acids, whereas the health-conscious group took significantly more supplements (Table 3). The tendency that people in the health-conscious group used more supplements was also found in the dietary patterns based on the modified PDI score, though with no significant difference (Appendix A).

A majority of people (90%) believed that their vegan diet was a varied diet, with a higher percentage in the health-conscious group (Table 3, Appendix A). Additionally, people in the convenience dietary pattern group had the feeling that they spent more money due to a vegan diet than those in the health-conscious group, with a significant difference in the groups based on the PDI score (Table 3, Appendix A). Over a third of all participants (36%) went to vegan-friendly restaurants or cafés at least weekly, while people in the health-conscious group visited cafés or restaurants less often (Table 3, Appendix A).

### 3.3. Physical Activity

The participants had a sitting time of 6.8 (SD: 3.1) h/day, 73% had ≥150 min of moderate–vigorous aerobic PA per week, 50% strength training ≥ 2 days/week, and 40% both, based on the guidelines for aerobic and strength exercises. When looking at gender differences, further results showed significant differences only in aerobic PA (female: 280 (240–330) min/week, male: 400 (300–500) min/week, nonbinary, not specified: 195 (60–300) min/week; *p* = 0.018).

Differences between the dietary patterns based on the PCA indicated that individuals in the health-conscious group were sitting about one hour less/day (Table 4).

Participants in the health-conscious group were significantly more likely to fulfill the aerobic PA guidelines and the recommendations for strength training (Figure 1). The fact that people who ate more consciously practiced significantly more PA was also confirmed by the dietary pattern analysis based on the PDI score (Appendix A).

Concerning yoga, 37% practiced yoga on a regular basis, and of those, 69% performed it ≥ 1–2 sessions/week, especially due to stress reduction (84%) and to increase flexibility (82%; multiple responses). Two-thirds (60%) specified an advanced level, and 48% practiced vinyasa yoga. Significantly more participants in the health-conscious group performed yoga on a regular basis (45% vs. 31%; *p* = 0.001).

### 3.4. Association between Vegan Dietary Patterns and PA

Vegans with a convenience dietary pattern had significantly higher odds of sitting more and not achieving the aerobic PA or strength training recommendations than those with a health-conscious dietary pattern (Table 5).

These associations remained significant even after adjusting for gender, employment, smoking status, and time being vegan (Table 5).

## 4. Discussion

Our study indicates that the quality of vegan diets among younger adults is heterogeneous, with a more health-conscious vs. a more convenience food-based vegan dietary pattern according to PCA. The identified convenience dietary pattern was characterized by higher consumption of processed fish and meat alternatives, vegan savory snacks, processed foods, sauces and condiments, cakes and biscuits, sweets and desserts, convenience meals and snacks, fruit juices/smoothies, and refined grains. By contrast, the health-conscious dietary pattern was characterized by higher consumption of vegetables, fruits, protein alternatives (e.g., tofu), dairy alternatives, potatoes, whole grains, vegetable oils and fats, and cooking with fresh ingredients and creating own recipes. Our results further showed that vegans with a health-conscious diet also practiced more PA. Specifically, sitting time was lower in the health-conscious dietary group and people in the group were more likely to achieve the recommendations from aerobic PA and the muscle-strengthening guidelines.

In the present study, we used different methods to classify the quality of the diet. The results consistently showed that higher diet quality, either as operationalized by PCA in a data-driven manner or by a predefined PDI score, is associated with a higher PA level. When using PCA, although it indicated the presence of five factors based on the screen plot and theoretical considerations, a two-factor solution was chosen. The cross-loadings indicated that a third factor could not be clearly separated from the other two. Therefore, and based on prior content knowledge, two factors or two dietary patterns were considered suitable. Concerning the PDI score, the variables were classified in accordance with Satija et al. [24]. When comparing the results of the PDI score method to those from a UK population in the study by Gallagher et al. (*n* = 129; 36% were 18–24 years), using cluster analysis, 27% were in the convenience group, 22% in the traditional, and 51% in the health-conscious group [10]. As there are different dietary patterns in the vegan diet and these patterns are also different in PA levels, a vegan diet per se cannot necessarily be equated with a healthy diet, as vegan diet quality must be addressed.

Interestingly, we found no significant differences in BMI or its classification between the convenience and the health-conscious groups based on the PCA. By comparing the BMI classification, 79% of our vegan study population showed a normal weight and 47% in the Austrian general population [25]. These results suggest that BMI in vegans is lower and the percentage of vegans with normal weight is higher compared to the general population. This observation has also been found in previous studies. For example, in the Adventist study (*n* = 22,434 men and 38,469 women) the BMI of vegans was 23.6 kg/m^2^ and in omnivores 28.8 kg/m^2^ [26]. Moreover, a meta-analysis by Dinu and colleagues found BMI values 1.7 kg/m^2^ lower in vegans in comparison to omnivores [27]. Therefore, it can be assumed that the low BMI values of our vegan study population might be the reason for the nonsignificant differences between the groups.

When looking at the reason for choosing a vegan diet, people adhering to a health-conscious vegan diet, according to PCA analyses, were more likely to state that health aspects were the reason for choosing a vegan diet. In the abovementioned study by Gallagher et al., 43% of the study participants stated that a combination of motives including health aspects, environmental and animal welfare reasons were the underlying reason for following a vegan diet. The other half of the participants gave other reasons (16% environmental protection and animal welfare, 22% animal welfare only, 6% environmental protection only, 5% health aspects and animal welfare, and 3% health aspects only) [10]. Our study showed similar results, i.e., 47% stated that health aspects, environmental protection and animal welfare aspects were the basis of their vegan diet, with 22% reporting environmental protection and animal welfare, 17% animal welfare only, 6% health aspects and animal welfare, 5% health aspects only, 3% health aspects and environmental protection, and 1% environmental protection only as reasons.

Concerning supplement use, we observed high percentages of vitamin B12 and vitamin D supplementation. These high rates are comparable to data from vegan health-care professionals attending a congress on plant-based nutrition (*n* = 213; vitamin B12: 100%, vitamin D: 73%) [28]. Apart from that, supplementation of n-3 fatty acids, iron, calcium, zinc, and selenium were even higher in the present data set (n-3 fatty acids: 22%; iron: 13%; calcium: 11%; zinc: 7%; selenium: 4%). Not all studies showed such a high rate of supplementation. For example, in an Australian analysis (*n* = 1530), 27% reported not taking vitamin B12 in the past three months, and of those supplementing, almost a quarter did not have an adequate supply [29]. Furthermore, in the already mentioned study by Gallagher and colleagues, the supplementation rate in UK participants was lower (vitamin B12: 68%; vitamin D: 42%; omega-3 fatty acids: 14%; iron: 26%; calcium: 19%; zinc: 15%; selenium: 8%) [10].

With regard to expenses for food and frequency of visits to vegan-friendly restaurants, a convenience dietary pattern with more UPFs and more restaurant visits seems to be more expensive. Many people assume that a vegan lifestyle is associated with higher costs, but a modeling study using regionally comparable food prices from 150 countries has shown the opposite [30]. The relative affordability was highest for vegetarian and vegan diets, which substituted whole grains and legumes for animal products in typical diets, and lowest for pescatarian diets, which prioritized fish and fruits, and vegetables over meat and other animal products [30].

Regarding PA, our results suggest that vegans are quite active, with a higher percentage achieving the aerobic PA and strength training recommendations compared to the general Austrian population, assessed in the Health Interview Survey 2019 (ATHIS) (vegans: 73% vs. general population: 44% and 50% vs. 28%) [25,31]. Therefore, the percentage of people reaching both the aerobic PA and strength training guidelines was about 18% higher in vegans (39% vs. 21%) [25,31]. In this context, it has to be kept in mind that with a mean age of 28 years, the participants in our study were younger than the Austrian general population. According to data from European Commission, 36% of men and 28% of women in Austria aged 18–29 years fulfill the PA recommendations [32]. Compared to other studies including people with a vegan diet, the present sample seemed to be very active. For example, compared to the cross-sectional study by Garcia-Maldonado et al., the vegan male and female participants in our study practiced a median of 58 and 28 min/week more vigorous PA and 75 and 60 min/week more moderate PA [33]. Subsequently, regarding the association between different vegan dietary patterns and PA, the health-conscious group had a significantly higher chance of fulfilling the PA recommendations. An association between a healthy diet and PA was also shown in other studies [34,35,36], even though—to our knowledge—it has not been demonstrated in people following a vegan diet. For example, the Framingham Heart Study (*n* = 2380) showed a relationship between greater cardiorespiratory fitness and healthy dietary pattern in middle-aged people [34]. Additionally, a systematic review and meta-analysis indicated that better adherence to the Mediterranean diet was associated with 2.3 higher odds of cardiorespiratory, 1.3 of musculoskeletal, and 1.4 of overall physical fitness than low adherence [35]. Considering that PA and lifestyle factors are not included as determinants in most of the studies concerning vegetarianism or veganism and body composition [36], and this can lead to bias. This is why the present study supports the demand of Fontes and colleagues [36], requesting that PA (including sedentary behavior) should necessarily be assessed in addition to nutritional data.

A high percentage of the sample performed yoga on a regular basis. In the general population of Germany, 5% practice yoga [37]. In an American population-based study (*n* = 1830 young adults), 42% reported practicing yoga regularly, and the most frequent reasons were the same as in our study (stress reduction, flexibility) [21]. In addition, a nationwide Australian survey (*n* = 28,695 women) demonstrated that vegans were more likely to perform yoga or meditation [38], and a cross-sectional U.S. study (*n* = 4307) showed that people practicing yoga had higher odds of following a vegetarian diet and had higher fruit and vegetable consumption [39]. These results are comparable to a previously published study demonstrating an association between healthier eating behavior (more servings of fruits and vegetables, fewer servings of sugar-sweetened beverages and snacks, and less frequent fast food consumption) and a higher chance of performing yoga [40].

As stated above, studies on the association between vegan dietary patterns and PA are broadly missing. Therefore, analyzing this relationship in the present study and the appropriately large sample can be considered a strength. Additionally, we used two different methods to categorize dietary patterns, with comparable results. However, limiting factors are that the respondents were recruited via social media, which may lead to selection bias, and data on diet and PA were based on self-reports.

Consequently, additional research is required in regard to vegan dietary patterns and physical fitness, as well as examining vegan dietary patterns in a longitudinal study or cohort study to assess their health effects. A complete dietary assessment with a focus on ultraprocessed foods, blood metabolite analysis, and body composition would also be necessary. It should also be noted here that dietary patterns and PA behavior were only surveyed in summer, which may have biased the results.

## 5. Conclusions

Due to the different vegan dietary patterns emerging from the present study and the resulting diet quality, a vegan diet exhibits some heterogeneity. This is also reflected in the differences in PA behavior in the dietary patterns, with vegans following a health-conscious dietary pattern practicing more PA compared to those with a convenience one. The results of our study suggest that especially the heterogeneity in the vegan diet and the relationship with lifestyle factors such as PA should be further examined.

## Figures and Tables

**Figure 1 nutrients-15-01847-f001:**
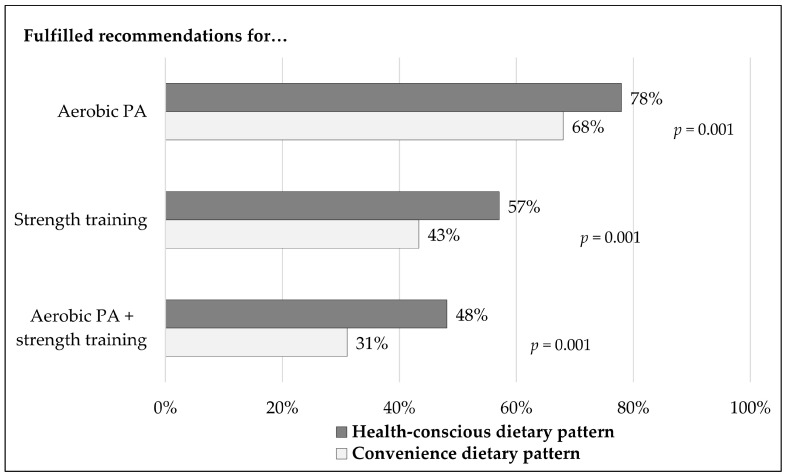
Percentages of people fulfilling the physical activity recommendations. Dietary patterns are based on the principal component analysis.

**Table 1 nutrients-15-01847-t001:** Dietary patterns derived from principal component analysis.

Factor Loadings	Dietary Patterns	
	Convenience	Health-Conscious
Processed fish and meat alternatives	0.701	-
Vegan savory snacks	0.650	-
Vegan processed foods	0.646	-
Vegan sauces and condiments	0.615	-
Vegan cakes and biscuits	0.540	-
Fruit juices/smoothies	0.519	-
Vegan sweets and desserts	0.476	-
Refined grains	0.472	-
Vegan convenience meals and snacks	0.307	-
Cooking with fresh ingredients	-	0.706
Vegetables	-	0.628
Protein alternatives	-	0.618
Creating own recipes	-	0.597
Fruits	-	0.579
Dairy alternatives	-	0.434
Potatoes	-	0.408
Whole grains	-	0.371
Vegetable oils and fats	-	0.314

**Table 2 nutrients-15-01847-t002:** Population characteristics divided by dietary patterns based on a principal component analysis.

		Dietary Patterns		
		Convenience (*n* = 271)	Health-Conscious (*n* = 243)	*p*
Time on vegan diet (years)		2.5 (2.5–3.0)	3.0 (3.0–4.0)	0.016
Age (years)		28.2 (7.4)	27.8 (8.2)	0.176
Gender	Female	81.5%	88.9%	0.025
	Male	14.8%	10.3%	
	Nonbinary, not specified	3.7%	0.8%	
Education	Compulsory school	10.7%	6.2%	0.186
	A-level	44.6%	47.7%	
	University	44.6%	46.1%	
Employment	Students or in training	33.9%	48.6%	0.002
	Employee or self-employed	62.0%	49.8%	
	Other	4.1%	1.6%	
Smoking status	Yes	10.3%	3.7%	0.009
	Occasionally	12.5%	10.3%	
	Former	17.3%	14.8%	
	Never	59.8%	71.2%	
BMI (kg/m^2^)		22.5 (4.1)	22.0 (3.2)	0.077
	Underweight	5.6%	6.7%	0.197
	Normal weight	76.8%	81.5%	
	Overweight	12.4%	9.7%	
	Obesity	5.2%	2.1%	

Metric data are given in mean (SD) or median (95% confidence interval), if the normal distribution was not given. Differences in groups were calculated using chi-squared was for categorical data and independent *t*-tests or Mann–Whitney U-test for metric data. *p* < 0.05 were considered statistically significant.

**Table 3 nutrients-15-01847-t003:** Reasons for adopting a vegan diet and taking nutritional supplements, divided by dietary patterns based on a principal component analysis.

		Dietary Patterns		
		Convenience	Health-Conscious	*p*
Reasons for veganism *	Health aspects	54.6%	64.9%	0.008
	Environment	71.7%	73.6%	
	Animal welfare	92.9%	89.3%	
	Religious beliefs	1.9%	0.0%	
	Weight reduction	1.5%	4.5%	
	Taste	0.7%	1.7%	
Supplements *	Vitamin B12	94.2%	94.3%	0.002
	Vitamin D	73.0%	76.5%	
	Omega-3 fatty acids	39.0%	48.3%	
	Iodine	14.1%	22.2%	
	Iron	45.6%	50.0%	
	Calcium	17.4%	21.3%	
	Zinc	25.7%	38.7%	
	Potassium	6.6%	10.0%	
	Vitamin K2	2.1%	2.6%	
	Selenium	17.4%	25.2%	
Vegan diet is varied ^#^	Yes	84.1%	95.5%	0.001
	No	5.9%	2.5%	
	Not sure	10.0%	2.1%	
Spending money due to vegan diet ^#^	More	33.6%	23.7%	0.085
	Less	13.7%	15.8%	
	No difference	46.1%	51.0%	
	Not sure	6.6%	9.5%	
Vegan-friendly cafés/restaurants visits (frequency)	Daily	1.8%	0.0%	0.001
	Weekly	42.7%	26.3%	
	Monthly	38.4%	47.7%	
	2–6 months	16.2%	22.6%	
	Never	1.1%	3.3%	

* Multiple responses; ^#^ personal opinion. Differences in groups were calculated using chi-squared tests for categorical data and independent *t*-tests or Mann–Whitney U-test for metric data. *p* < 0.05 was considered statistically significant.

**Table 4 nutrients-15-01847-t004:** Physical activity divided by dietary patterns based on a principal component analysis.

	Dietary Patterns		
	Convenience	Health-Conscious	*p*
Sitting time (hours/day)	7.2 (3.2)	6.3 (2.9)	0.001
Aerobic PA (min/week)	240 (190–300)	360 (330–420)	0.002
Moderate (min/week)	120 (190–180)	120 (120–165)	0.676
Vigorous (min/week)	93 (80–120)	120 (120–180)	0.001
Strength training			
Never/<1 day/week	39.5%	25.8%	0.003
1 day/week	17.1%	17.1%	
≥2 days/week	43.3%	57.1%	

Metric data are given in mean (SD) or median (95% CI), if the normal distribution was not given. Differences in groups were calculated using chi-squared tests for categorical data and independent *t*-tests or Mann-Whitney *U*-test for metric data. *p* < 0.05 was considered statistically significant.

**Table 5 nutrients-15-01847-t005:** Logistic regression analyses of associations between dietary patterns and PA.

		Dietary Patterns (Ref.: Health-Conscious)						
	Crude				Adjusted *			
	OR	95% CI	*p*	R^2^	OR	95% CI	*p*	R^2^
Sitting time (hours)	1.10	1.04–1.17	0.001	0.028	1.10	1.04–1.18	0.002	0.097
Not fulfilling aerobic PA recommendations	1.72	1.13–2.61	0.011	0.019	1.81	1.18–2.79	0.007	0.093
Not fulfilling strength recommendations	1.74	1.22–2.48	0.002	0.025	1.81	1.26–2.61	0.002	0.101

Results from the logistic regression analyses (dependent variables: health-conscious as the reference group) are presented as odds ratios (ORs) with their 95% confidence intervals (95% CIs). * Results were adjusted for gender, employment, smoking status, and time being vegan. R^2^: Nagelkerke measure of goodness of fit; *p* < 0.05 considered statistically significant.

## Data Availability

The data presented in this study are available on request from the corresponding author.

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
