# Peer review of "The Association between Vegan Dietary Patterns and Physical Activity—A Cross-Sectional Online Survey"

_nutrients, 2023, doi:10.3390/nu15081847_

Round 1

Reviewer 1 Report

After reviewing the manuscript content, the following observations are transferred to the authors:

ABSTRACT:

In the keywords, it could be specified what type of physical activity or exercise has been involved in the study.

INTRODUCTION:

- The literature review related to diet is sufficiently broad. However, the one referring to physical activity is excessively brief (lines 71-79), without delving too deeply into the type of activity or exercises program carried out.

MATERIALS AND METHODS:

For recruitment, how the Facebook groups and Instagram accounts contacted were identified to participate in the study? Why were these groups/accounts selected, and not others? Contact criteria?

- Instead of using the BMI, which is really a weight-height ratio, it would have been more appropriate to record values ​​for % fat mass and % lean mass. Knowing that the data has already been collected, this is a recommendation for future studies.

RESULTS:

- The tables in this section do not show significant differences. They should be marked with a symbol (eg "*"), and the significance level should be described in the legend.

Author Response

Thank you for doing the review of our manuscript. The answers to our questions are in the appendix.

Reviewer 1

Location in the tracked version

Thank you very much for having the opportunity to review the paper. The introduction, presentation, results and discussion are well built. However, some questions need to be resolved:

We kindly thank you for your comments. We have tried to answer them as well and detailed as possible.

1. In Abstract, it is read: “...Different dietary patterns were compiled through principal component analysis”. Also, in the Section 2.4 it is read: “... First, food and beverage items were combined and collapsed into 18 food groups based on Gallagher et al. [10] for applying factor analysis. Both Bartlett's test (chi-square = 1961.4, p < .001) and the Kaiser-Meyer-Olkin Measure of Sampling Adequacy (KMO = .805) indicate that the variables are suitable for factor analysis. Therefore, the dietary patterns were assembled from the 18 main food groups and eating behavior variables by principal component analysis (PCA) …”

Principal Component Analysis (PCA) and Factor Analysis are completely different statistical techniques. It should be made more clear in the methodology section how these two distinct methodologies were applied, whether in a combination, being PCA the estimation method for the factor analysis, or as totally separated (and sequential analysis) with different outcomes.

Thank you for bringing this to our attention, and we excuse ourselves for not being clear about it and for using the term factor analysis, but we meant PCA instead.

We have used two methodologies: the a posteriori and the a priori approach. The a posteriori method is using multivariate statistical techniques, such as PCA (a data-driven approach). The a priori approach is based on nutrition guidelines or recommendations, uses predefined dietary patterns, and measures the level of adherence to these patterns.

We have revised this section accordingly.

“For deriving dietary patterns, we have used an a posteriori and a priori approach. First, food and beverage items were combined and collapsed into 18 food groups based on Gallagher et al. [10] for applying principal component analysis (PCA; a posteriori approach). Both Bartlett's test (chi-square = 1961.4, p < .001) and the Kaiser-Meyer-Olkin Measure of Sampling Adequacy (KMO = .805) indicate that the variables are suitable for PCA.”

P. 4

2. In the Section 2.4 it is read: “... Therefore, the dietary patterns were assembled from the 18 main food groups and eating behavior variables by principal component analysis (PCA), using Oblimin and Varimax rotations [22].”

Rotation algorithms are usually used in Factor Analysis to improve the interpretation of results, with each rotation dealing with different aspects of analysis. Therefore, only a single rotation is usually used. Please, confirm if two rotations were used in this manuscript and explain the motivations for each one.

Again, thank you so much for bringing this to our attention. We only used the varimax rotation. So only a single rotation. Sorry for not being precise here.

“Therefore, the dietary patterns were assembled from the 18 main food groups and eating behavior variables by PCA, using varimax rotation [22].”

P. 4

3. Lines 172-182 it is read: “Following Satija et al. [24] the variables were classified into quintiles and assigned positive for healthful plant-based diet index (hPDI) or reverse scores for unhealthful plant-based diet index (uPDI). Participants above the highest quintile of a food group received a positive score of five, while those below the lowest quintile received a negative score of one. This scoring pattern was reversed. With this analysis, we divided the people into three groups (1st tertile= convenience; 2nd tertile= traditional; 3rd tertile=health-conscious dietary pattern). The results of the second method are shown in the supplementary tables.”

Unfortunately, I don’t understand this methodology. My understanding was: Consider an individual who has a healthful plant-based diet, staying above the highest quintile, they receives a score of 5. In the second step, this scoring pattern is reversed, and they receives a score of -1. With this reversal, they belongs to the 1st tertile (convenience group). Is there any mistake in my reasoning? Could the authors better explain what the scoring pattern inversion means? 

Please excuse us for not describing this in more detail. We have revised this section accordingly.

In addition to data-driven dietary patterns, we assessed diet quality by the modified plant-based diet index (PDI; a priori approach) [23]. For the PDI, we used the 18 food groups divided into healthy plant-based foods & dietary behavior (cooking with fresh ingredients, vegetables, protein alternatives, creating own recipes, fruits, dairy alternatives, potatoes, whole grains, and vegetable oils & fats) and less healthy plant-based foods & dietary behavior (processed fish & meat alternatives, vegan savory snacks, vegan processed foods, vegan sauces & condiments, vegan cakes & biscuits, fruit juices/smoothies, vegan sweets & desserts, refined grains, and vegan convenience meals & snacks). Following Satija et al. [24] the variables (healthy and less healthy plant-based foods) were classified into quintiles and assigned positive for the healthful plant-based diet index (hPDI) or reverse scores for the unhealthful plant-based diet index (uPDI). Participants above the highest quintile of a healthy plant-based food group received a positive score of five, while those below the lowest quintile received a score of one. This scoring pattern was reversed when applying to the less healthy plant-based food groups, e.g. the highest quintile received a score of 1 and the lowest quintile a score of 5. With this analysis, we divided the people into three groups: 1st tertile= convenience; 2nd tertile= traditional; 3rd tertile= health-conscious dietary pattern. The results of the a priori method are shown in the supplementary tables.”

P.5

4. Figure 1: It is necessary to increase the resolution of this figure.

We have changed the resolution of the figure, thank you.

Figure 1 (P. 9)

Supplementary Figure 1 (P. 4)

5. Figure 1: All analysis obtained from division according to tertiles (three groups considered) are presented in supplementary material. Why is this graph presented inside the manuscript (not in supplementary material)?

Based on your comment, we have moved Figure 1B (dietary patterns are based on modified PDI score) to the supplementary material. This figure is now named Supplementary Figure 1.

Figure 1 (P. 9)

Supplementary Figure 1 (P. 4)

6. When proposing a multivariate logistic analysis, it is necessary to assess the goodness of fit through measures such as AUC, sensitivity, specificity, error rate among others. However, the authors did not present any analysis of the model's goodness of fit.

We would like to thank you for this comment. We have added the Nagelkerke R2 to the adjusted models. Furthermore, we have mentioned it in the method section.

"To interpret the goodness of fit, we presented the Nagelkerke R Square."

Table 5

P.5

Reviewer 2 Report

Thank you very much for having the opportunity to review the paper. The introduction, presentation, results and discussion are well built. However, some questions need to be resolved:

1. In Abstract, it is read: “...Different dietary patterns were compiled through principal component analysis”. Also, in the Section 2.4 it is read: “... First, food and beverage items were combined and collapsed into 18 food groups based on Gallagher et al. [10] for applying factor analysis. Both Bartlett's test (chi-square = 1961.4, p < .001) and the Kaiser-Meyer-Olkin Measure of Sampling Adequacy (KMO = .805) indicate that the variables are suitable for factor analysis. Therefore, the dietary patterns were assembled from the 18 main food groups and eating behavior variables by principal component analysis (PCA) …”

Principal Component Analysis (PCA) and Factor Analysis are completely different statistical techniques. It should be made more clear in the methodology section how these two distinct methodologies were applied, whether in a combination, being PCA the estimation method for the factor analysis, or as totally separated (and sequential analysis) with different outcomes.

2. In the Section 2.4 it is read: “... Therefore, the dietary patterns were assembled from the 18 main food groups and eating behavior variables by principal component analysis (PCA), using Oblimin and Varimax rotations [22].”

Rotation algorithms are usually used in Factor Analysis to improve the interpretation of results, with each rotation dealing with different aspects of analysis. Therefore, only a single rotation is usually used. Please, confirm if two rotations were used in this manuscript and explain the motivations for each one.

3. Lines 172-182 it is read: “Following Satija et al. [24] the variables were classified into quintiles and assigned positive for healthful plant-based diet index (hPDI) or reverse scores for unhealthful plant-based diet index (uPDI). Participants above the highest quintile of a food group received a positive score of five, while those below the lowest quintile received a negative score of one. This scoring pattern was reversed. With this analysis, we divided the people into three groups (1st tertile= convenience; 2nd tertile= traditional; 3rd tertile=health-conscious dietary pattern). The results of the second method are shown in the supplementary tables.”

Unfortunately, I don’t understand this methodology. My understanding was: Consider an individual who has a healthful plant-based diet, staying above the highest quintile, they receives a score of 5. In the second step, this scoring pattern is reversed, and they receives a score of -1. With this reversal, they belongs to the 1st tertile (convenience group). Is there any mistake in my reasoning? Could the authors better explain what the scoring pattern inversion means? 

4. Figure 1: It is necessary to increase the resolution of this figure.

5. Figure 1: All analysis obtained from division according to tertiles (three groups considered) are presented in supplementary material. Why is this graph presented inside the manuscript (not in supplementary material)?

6. When proposing a multivariate logistic analysis, it is necessary to assess the goodness of fit through measures such as AUC, sensitivity, specificity, error rate among others. However, the authors did not present any analysis of the model's goodness of fit.

Author Response

Thank you for doing the review of our manuscript. The answers to the questions are in the appendix.

Reviewer 2

Location in the tracked version

Interesting, well-documented research on “The association between vegan dietary patterns and physical activity – a cross-sectional online survey”.

Thank you for your recognition as well as for investing your time in the review. We really appreciate your comments.

Conducting the research only during one season of the year, i.e. the summer of 2022, may raise doubts (respondents were asked about their eating habits only in the last month). Physical activity as well as nutrition during the summer may differ from the whole year.

As we totally agree with this comment, we have added the following sentence to the limitations:

It should also be noted that dietary patterns and the PA behavior were only surveyed in summer, which may have biased the results.“

P. 12

The novelty of the research is not sufficiently described in the introduction.

Based on this comment, we have reformulated the following sentences in the introduction section:

“Due to the heterogeneity in vegan diets and as studies on the association between vegan dietary patterns and PA are broadly missing, the aim of the paper was to 1) assess dietary patterns in people who adopted a vegan diet; and 2) analyze the association between different vegan dietary patterns and the extent of PA.”

Furthermore, it is stated in the introduction that:

“Overall, few published studies addressed the aspect of vegan diet heterogeneity, but those suggest that vegan diet quality varies significantly.”

P. 2

P. 2

The discussion contains repetitions of the results.

We agree with you and have made the following changes

We have deleted the following sentences:

·       „In our study, 30% were in the convenience group, 33% in the traditional, and 37 % in the health-conscious group.“

·       „In our study, when looking at the group differences, vitamin B12 and D supplementation were equal between the groups, but more individuals in the health-conscious group used iodine, iron, calcium, zinc, potassium, vitamin K2, and selenium supplementation.“

·       „The main reason for choosing a vegan diet was animal welfare, followed by environmental and health aspects“

We have also reworded the sentence to

·       „With regard to expenses for food and the frequency of visits in vegan-friendly restaurants, especially a convenient dietary pattern with more UPFs and more restaurant visits seems to be more expensive.“

P. 10- P. 12

Reviewer 3 Report

Interesting, well-documented research on “The association between vegan dietary patterns and physical activity – a cross-sectional online survey”.

Conducting the research only during one season of the year, i.e. the summer of 2022, may raise doubts (respondents were asked about their eating habits only in the last month).

Physical activity as well as nutrition during the summer may differ from the whole year.

The novelty of the research is not sufficiently described in the introduction.

The discussion contains repetitions of the results.

Author Response

Thank you for doing the review of our manuscript. The answers to the questions are in the appendix.

Reviewer 3

Location in the tracked version

The methods applied are not very precise and bias can be present. Anyway as the idea is interesting and in the discussion authors already point the few fragilities of the study I think it can be published as an awareness to future studies.

Thank you for the review and this comment. We appreciate your investing time in this review. Based on the comments of the other reviewers, we have further expanded the limitations of the study.

Now the following is mentioned:

“Consequently, additional research is required in regard to vegan dietary patterns and physical fitness as well as examining vegan dietary patterns in a longitudinal study or cohort study to assess their health effects. Additionally, a complete dietary assessment with a focus on ultra-processed foods and blood metabolite analysis and also body composition would be necessary. It should also be noted here that the dietary patterns and the PA behavior were only surveyed in summer, which may have biased the results.”

P. 12

Last paragraph

Reviewer 4 Report

The methods applied are not very precise and bias can be present. Anyway as the idea is interesting and in the discussion authors already point the few fragilities of the study I think it can be published as an awareness to future studies.

Author Response

Thank you for doing the review of our manuscript. The answers to the questions are in the appendix.

Reviewer 4

Location in the tracked version

ABSTRACT: In the keywords, it could be specified what type of physical activity or exercise has been involved in the study.

Thank you, we have added the following keywords:

“strength training, yoga, aerobic physical activity”

P. 1

INTRODUCTION: The literature review related to diet is sufficiently broad. However, the one referring to physical activity is excessively brief (lines 71-79), without delving too deeply into the type of activity or exercises program carried out.

Thank you very much for this comment. We completely agree with you that there is an imbalance between diet and exercise in the introduction section. However, we believe that the background on nutrition needs has to be explained in more detail. Additionally, after a closer check, we are convinced that all the necessary literature concerning PA for understanding the present paper is mentioned. Therefore, we have not made any changes in this respect.

MATERIALS AND METHODS: For recruitment, how the Facebook groups and Instagram accounts contacted were identified to participate in the study? Why were these groups/accounts selected, and not others? Contact criteria?

There were no clear criteria in this respect. The groups were chosen on the recommendations of vegans, who were active on these platforms. To clarify this, the following sentence has been added to the method section:

„These groups were chosen based on the recommendations of vegans, who were active on these platforms.“

P. 2 - P.3

Instead of using the BMI, which is really a weight-height ratio, it would have been more appropriate to record values ​​for % fat mass and % lean mass. Knowing that the data has already been collected, this is a recommendation for future studies.

We totally agree with the reviewer that it would have been very interesting to present %fat and %lean mass. However, since the present work was an online survey, no data is available in this respect. To clarify this, the limitations were extended as follows.

„Additionally, a complete dietary assessment with a focus on ultra-processed foods, blood metabolite analysis, and body composition would also be necessary.“

P. 12

last paragraph

RESULTS: The tables in this section do not show significant differences. They should be marked with a symbol (eg "*"), and the significance level should be described in the legend.

Thank you for this comment. We have deliberately decided against highlighting significant values, in order to prevent over-interpretation, for the following reasons:

-          Stahel writes in his publication (https://journals.plos.org/plosone/article?id=10.1371/journal.pone.0252991): A classification of results that goes beyond a simple distinction like "significant / non-significant" is proposed.

-          Moreover, our results are an explorative design in which we did not apply a Bonferroni correction for multiple testing.

Nevertheless, based on this comment, we have mentioned the significance level in the legend.

Table 2-5

Supplementary Table 1-3
